# Understanding and Targeting Metabolic Vulnerabilities in Acute Myeloid Leukemia: An Updated Comprehensive Review

**DOI:** 10.3390/cancers17081355

**Published:** 2025-04-18

**Authors:** Sridevi Addanki, Lana Kim, Alexandra Stevens

**Affiliations:** Division of Pediatric Hematology/Oncology, Department of Pediatrics, Baylor College of Medicine, Houston, TX 77030, USA

**Keywords:** acute myeloid leukemia, hematopoietic stem cells, leukemic stem cells, glycolysis, oxidative phosphorylation, metabolism, atovoquone, venetoclax, azactidine

## Abstract

Acute Myeloid Leukemia (AML) is a type of blood cancer that grows quickly and is difficult to treat. One way to fight AML is by understanding how its cells generate and use energy differently from normal blood cells. Unlike healthy blood stem cells, AML cells have unique energy production methods that help them grow and survive. This review explores these metabolic differences, identifying weak points that could be targeted with new treatments. By studying how AML cells rely on specific energy pathways, scientists can develop drugs that disrupt cancer growth while sparing normal cells. However, AML is a complex and varied disease, making treatment challenging. Continued research is needed to improve therapies and develop more precise treatments that specifically target AML’s energy use without harming healthy blood cells.

## 1. Introduction

Leukemia is a malignant clonal neoplastic disease of the hematopoietic stem cells (HSCs) characterized by abnormal hyperplastic hematopoietic progenitor cells that accumulate in bone marrow or other blood-associated tissues. These transformed cells differ from the normal hematopoietic cell functions by their uncontrolled proliferation, compromised differentiation potential, and altered resistance to apoptosis [1]. Acute Myeloid Leukemia (AML) is the most common type of leukemia in adults and has a poor prognosis, with its incidence increasing with age [2]. Young patients with AML tend to have better outcomes compared to older adults, who generally face a poorer prognosis [3]. This discrepancy can be attributed to elderly patients often being unable to tolerate intense chemotherapy, existing clinical comorbidities, and mutations that lead to resistance to systemic therapies [4,5].

AML is an aggressive and heterogeneous disease characterized by the rapid proliferation of abnormal, immature progenitor cells of the myeloid lineage [6]. The rapid proliferation of myeloblasts in the bone marrow can lead to decreased normal hematopoiesis, neutropenia, and increased susceptibility to infectious agents [7].

Multiple factors contribute to the development of AML, including medical comorbidities, exposure to chemical toxins, previous chemotherapy or radiation treatment, genetic predispositions, prior myelodysplastic syndromes, and the overall aging population [8]. Chromosomal translocations and genetic mutations have also been attributed to the development of AML [9]. AML is marked by heterogeneous cell phenotypes and molecular events, so abnormalities in cytogenetics, somatic variants, and cell surface markers are used for diagnostic purposes and disease monitoring in a patient-specific manner [10]. Adults diagnosed with AML are risk-classified by cytogenetic and mutational events, and further risk stratification can be achieved by determining the minimal residual disease (MRD) after initiation of chemotherapy [11,12].

Until recently, the best available treatment option for patients newly diagnosed with AML included intensive cytotoxic chemotherapy with daily infusions of cytarabine (Ara-C) and high doses of anthracycline [13]. Cytarabine is a nucleoside analogue and is commonly used in combination with anthracycline daunorubicin for induction therapy. This backbone treatment, known as the “7 + 3” regimen, has been used for many decades and has demonstrated effectiveness, particularly in younger patients with a de novo diagnosis, yielding 5-year survival rates of 40–50% [14,15]. Patients who achieve remission after induction treatment may be considered for post-remission consolidation therapy, which typically includes cytarabine as a backbone agent [14,16]. In many cases, hematopoietic stem cell transplantation (HSCT) is also considered with the intent of achieving a cure [17].

Over the last few decades, an increased understanding of the genetic alterations driving AML has led to the development of novel therapeutics for AML. Thus, in many cases, it is now feasible to target the molecular and epigenetic characteristics of the disease and several of these agents are now FDA-approved [18,19]. Patients harboring mutations in FMS-related tyrosine kinase 3 gene (FLT3) internal tandem duplication (ITD subtype) or a point mutation in the tyrosine kinase domain (TKD subtype) can be targeted with tyrosine kinase inhibitors such as midostaurin or gilteritinib [20,21]. Isocitrate dehydrogenase mutations can be targeted with ivosidenib (IDH1) and enasidenib (IDH2) to remove the differentiation blockade that results from these mutations [22,23]. The hedgehog pathway inhibitor, Glasdegib, is being used along with low-dose Ara-C for the treatment of AML [24,25]. Glasdegib has been approved by the FDA in combination with low-dose cytarabine for newly diagnosed AML patients aged 75 and older or those with comorbidities that prevent the use of intensive induction chemotherapy. AML cells expressing CD33 can be targeted with gemtuzumab ozagamicin, an antibody drug conjugate delivering a calicheamicin payload [26]. Mylotarg (gemtuzumab ozagamicin) received FDA approval in 2000 for the treatment of patients with relapsed or refractory AML who are positive for CD33 [27].

Despite advances in our understanding of the biology of AML and in some targeted treatment modalities, AML remains a leading cause of cancer-related mortality, with five-year survival rates hovering around 32% for patients over the age of 60 [2,28]. The current landscape of AML treatment faces several formidable challenges, including the high toxicity associated with conventional chemotherapies, limited efficacy in eradicating leukemic stem cells, and the emergence of resistant AML clones in part due to the sub clonal nature of the disease. These challenges underscore the urgent need for novel therapeutic strategies that can target AML more selectively and effectively [29].

Recently, Venetoclax, a selective Bcl-2 inhibitor and a key player in mitochondrial pathways of cell survival, has shown great promise as an adjunctive therapy for AML. Venetoclax can be combined with hypomethylating agents such as azacytidine and decitabine or low-dose Ara-C as a frontline AML treatment for patients unfit for intensive chemotherapy [30,31]. Venetoclax can also be administered with intensive chemotherapy and has shown impressive results for achievement of both MRD negative complete remissions and overall event free survival [30]. Clinical data demonstrating the efficacy of venetoclax-based therapies in AML highlight the significance of targeting mitochondrial metabolism as a relevant strategy in leukemia [32,33,34,35].

Over the last decade, there has been growing interest in targeting the specific metabolic needs of cancer cells. This is based on the recognition that metabolic reprogramming is a hallmark of cancer, including in AML. Metabolic rewiring supports tumorigenesis and tumor progression by increased production and usage of essential nutrients. Cancer cells have altered metabolic pathways to meet the enhanced demand for energy, reducing equivalents, and biosynthetic precursors. Metabolic reprogramming in AML is characterized by alterations in cellular processes such as glycolysis, oxidative phosphorylation, and amino acid metabolism, all of which support the rapid growth and survival of leukemic cells under various stress conditions [36].

The rationale for targeting metabolic vulnerabilities in AML is multi-faceted. First, it provides an opportunity to disrupt the energy supply and biosynthetic machinery that is essential for the proliferation and survival of AML cells. Second, it offers a potential therapeutic strategy to overcome drug resistance and reduce toxicity by sparing normal cells, which have different metabolic requirements. Lastly, metabolic targeting can be tailored based on the unique metabolic dependencies of individual AML subtypes, paving the way for precision medicine approaches [36,37].

This review aims to dissect what is known about the metabolic characteristics of AML in comparison to normal hematopoiesis, delineate the altered metabolic pathways contributing to AML pathogenesis, and explore the therapeutic potential of targeting these metabolic vulnerabilities. By doing so, we aspire to highlight the emerging opportunities and challenges in exploiting metabolic reprogramming as a therapeutic strategy against AML.

## 2. Metabolic Characteristics of Normal Hematopoietic Cells and Leukemic Cells

### 2.1. Hematopoietic Stem Cell Metabolic Characteristics

The hematopoietic system is a paradigm of cellular hierarchy and differentiation, where HSCs residing at the apex maintain blood cell production throughout life via a finely tuned process known as hematopoiesis. Adult HSCs exist in different tissues and are present throughout an individual’s lifetime in order to maintain homeostasis of the hematopoietic tissue [38]. They ensure blood regeneration during damage and a physiologic turnover under steady-state conditions. Recent studies have shown that HSCs have unique metabolic characteristics. These properties enable them to maintain metabolic dormancy and a low cycling status in order to preserve stemness [39]. The maintenance of quiescence or differentiation is also determined by the location where adult HSCs reside, and this process is influenced by both cell-intrinsic and cell-extrinsic microenvironmental factors [40]. HSCs reside in particular bone marrow (BM) niches. For example, non-replicating HSCs are found in perivascular niches, which have a hypoxic microenvironment. This environment helps them maintain an energy demand state conducive to anaerobic glycolysis, ensuring stemness and quiescence [40]. The BM, or the hematopoietic microenvironment, acts as a niche for HSC and is a dynamic tissue composed of cellular and non-cellular elements [41]. The cellular components supporting HSCs include osteoblasts, osteoclasts, adipocytes, mesenchymal stromal cells, non-myelinating Schwann cells, sympathetic neuronal cells, endothelial cells, and tumor-associated immune cells. The non-cellular components mainly consist of soluble factors and the extracellular matrix [42,43].

HSCs possess the ability to self-renew and differentiate into multiple lineages, ensuring the continuous repopulation of mature hematopoietic tissue. The main functions of HSCs—self-renewal and differentiation—are tightly regulated by intrinsic mechanisms, including transcriptional and epigenetic regulation. Additionally, extrinsic signals from the bone marrow microenvironment also play a crucial role in these processes [40].

When HSCs divide, they face two possible fates: they can either undergo self-renewal to produce new HSCs or differentiate into cells that mature into committed cell types. Depending on the status of the daughter cells, HSCs can undergo asymmetric or symmetric cell division. Asymmetric division is essential for maintaining the HSC pool, as one daughter cell remains a stem cell while the other differentiates. In contrast, during symmetric cell division, both daughter cells retain their stem cell characteristics [44].

The metabolic dependencies specific to the cell state observed in leukemia development are also present during normal hematopoiesis [45]. The changes occurring in glucose utilization differently sensitize the stem cells and the progenitor cells. Energy metabolism in normal HSCs contrasts with more differentiated progenitor cells, which predominantly rely on glycolysis. Under steady-state conditions, HSCs exhibit a unique metabolic phenotype characterized by low rates of glycolysis and a reliance on OXPHOS for energy production. This metabolic setup is critical for maintaining the quiescence, self-renewal, and long-term regenerative capacity of HSCs [46]. During hematopoiesis, HSCs primarily utilize glycolysis and OXPHOS to a lesser extent when they are in a dormant state. HSC quiescence is dependent on pyruvate dehydrogenase kinase (PDK)-mediated glycolysis, which supports this process [47]. Also, there are transcriptional changes that promote HSC quiescence. One example of this is the expression of transcription factor MEIS1, promoting the expression of HIF-1α [47,48]. The low basal OXPHOS of healthy dormant HSCs is attributed in part to the expression of steroid receptor coactivator 3 [45]. However, there is a switch to a high OXPHOS phenotype during the differentiation process [46,49]. Thus, the metabolic state of HSCs is not static; it is dynamically regulated by environmental cues, such as nutrient availability and oxygen levels, thereby linking metabolic activity to hematopoietic demand [48]. During differentiation, HSCs use mitochondrial OXPHOS due to greater energy needs, and anaerobic glycolysis is used during more dormant periods [50]. Anaerobic glycolysis helps the HSCs meet the lower basal energy demands and helps them survive in the hypoxic BM microenvironment and maintain cellular integrity [39].

For routine energy requirements, HSCs avoid aerobic mitochondrial OXPHOS as they generate deleterious reactive oxygen species (ROS). ROS generation can lead to mitochondrial DNA mutations, which, in turn, inhibit HSC function [51]. ROS functions as a mediator for the signaling switch between metabolism and stem cell fate by decreasing the capacity of HSCs to repopulate [52,53]. ROS-high HSCs have decreased self-renewal potential and increased myeloid differentiation potential, while ROS-low HSCs have high self-renewal capacity. One method of protection from generated ROS that HSCs utilize is the transfer of excess ROS to the surrounding stromal population.

Despite the growing evidence demonstrating HSCs being glycolytic with decreased OXPHOS and ROS, studies have shown that quiescent HSCs have higher mitochondrial content and high levels of large lysosomes, both of which are associated with high OXPHOS [49,54]. Thus, there remains a critical need for accurate assessments of dynamic biological processes, such as the determination of mitochondrial membrane potential (MMP) in hematopoietic stem and progenitor cells [55]. It was observed in the last several years that HSCs have unique metabolic properties that promote a metabolically dormant and low cyclic state that enables stemness [39]. Mitochondrial distribution during stem cell division plays a role in determining the HSC fate. During asymmetric stem cell division, the daughter cells receiving fewer mitochondria maintain stem cell characteristics, while those receiving more mitochondria go on to differentiate. The tight regulation of metabolism in HSCs is crucial for preventing leukemogenesis. Disruptions in metabolic pathways, either through genetic mutations or external stressors, can lead to the unchecked proliferation and survival of abnormal cells, setting the stage for the development of leukemia [46,56].

### 2.2. Leukemic Cells and Leukemic Stem Cell Metabolic Characteristics

AML cells differ from normal hematopoietic progenitors with distinct mitochondrial signatures such as higher copy numbers of mitochondrial DNA (mtDNA), mitochondrial mass, and increased mitochondrial biogenesis. They also exhibit paradoxical dependency on OXPHOS and altered translation, which helps in cancer cell survival [57,58]. The high mitochondrial mass in AML cells can be attributed to mitochondrial transfer. In an elegantly conducted experiment in which a co-culture system mimicked the BM niche, it was shown that AML cells increased their mitochondrial mass by more than 14% by transferring functional mitochondria from BM stromal cells [59]. Moreover, the AML cells harboring transferred mitochondria show less chemotherapy-induced mitochondrial depolarization and thus increased survival [59]. In relapsed AML cells, there is an increase in the mitochondrial ribosomal proteins as well as the subunits of the electron transport chain, suggesting rewired energy metabolism [60].

AML patients have rapidly dividing and highly proliferative myeloblasts and also a second population of leukemic cells called LSCs. During the treatment of leukemia, most cells are eliminated, but some therapy-resistant cells, including LSCs, survive. Various mechanisms have been identified as contributing to the persistence of LSCs and treatment-resistant cells, with cellular metabolic reprogramming being a key adaptation. Given the importance of energy metabolism in HSCs, the metabolic rewiring in LSCs that differentiates them from normal HSCs may demonstrate a therapeutic vulnerability.

At first, LSCs were identified as leukemia-initiating cells engrafting in severely immunodeficient mice [61]. As their transcriptional and epigenetic signatures were identified, it became apparent that LSCs are independent of mutations and are linked to significantly inferior clinical outcomes [62,63]. Thus, LSCs are an important cell population in leukemia, leading to disease persistence. Additionally, a distinct population of cells called pre-LSCs, which harbor early mutations of the disease, were discovered and shown to be reservoirs of disease relapse [64]. In a seminal manuscript, Rothenberg and colleagues described the persistence of pre-leukemic clones during first remission and the association with relapse risk in AML [64].

Stephan Paget’s “Seed and Soil” hypothesis suggests that the BM microenvironment serves as the soil for the maintenance, self-renewal, and differentiation properties of the stem cells [65]. LSCs rely not only on intrinsic genetic programming but also on extrinsic cues from the BM microenvironment to sustain their self-renewal and therapy evasion. Thus, the BM niche can modulate LSC metabolism by supplying nutrients, altering redox balance, or inducing mitochondrial adaptations that promote LSC quiescence and survival. The metabolic crosstalk between LSCs and the BM niche is a key determinant of disease progression and treatment resistance. Targeting this metabolic symbiosis offers a compelling strategy to disrupt the protective ‘soil’ and sensitize LSCs to therapy.

It is important to note that while therapy-resistant cells are not all LSCs, LSCs are indeed resistant to treatment. Therapy-resistant cells are a clonal population that is selected during treatment. Acquired or selected mutations can make the cells resistant to targeted agents such as TKI, the BCL-2 inhibitor venetoclax, or chemotherapeutics [35,66,67,68,69]. Targeting mutations in AML is challenging due to the rapidly evolving nature of the disease. Additionally, disease monitoring using advanced deep sequencing methodologies is limited. Therefore, it is crucial to develop strategies to target persistent treatment-resistant cells that do not rely solely on targeting mutations. Leukemogenesis involves extensive metabolic rewiring, and targeting OXPHOS is a relevant clinical strategy due to its significant role in LSC maintenance.

Understanding the metabolic changes associated with LSCs is important, especially considering the significance of HSCs and energy metabolism. Many studies demonstrated the importance of OXPHOS and mitochondria transfer in AML LSCs [59,70,71,72]. One study found that LSCs are metabolically less active than the bulk cell population [73]. However, unlike other cells, LSCs do not rely on glycolysis and rely more on OXPHOS for their energy needs [73]. A proteomic study comparing AML LSCs with HSCs and AML blasts found that LSCs have upregulated electron transport chain (ETC) complexes I and V [74]. Alongside these changes in ETC complexes, LSCs also have altered mitochondrial regulation [75]. Multiple studies indicate that the mitochondrial content in AML LSCs is lower than in both HSCs and bulk AML cells [70,75]. It was revealed that the survival and leukemia-initiating abilities of LSCs rely on mitochondrial fission through the mitochondrial fission protein 1 (FIS1) and subsequent autophagy [75]. FIS1 also plays a role in the clearance of dysfunctional mitochondria, which possibly accounts for the lower mitochondrial content in AML LSCs compared to HSCs [70]. The expression of F1S1 is correlated to poor outcomes in AML and is an important LSC biomarker [76]. MTCH2 plays a role in the transport of pyruvate from the cytoplasm into the mitochondria, and inhibiting MTCH2 led to the accumulation of pyruvate in the cytoplasm and an increase in pyruvate dehydrogenase (PDH) levels in the nucleus [77]. It is linked to the survival and differentiation of AML LSCs and plays a significant role in the differentiation of both embryonic stem cells and HSCs [78,79]. An additional study demonstrated the role of mitochondrial transport in protein folding occurring in the mitochondrial intermembrane space, with several genes involved being overexpressed in LSCs [80]. These studies highlight the significance of mitochondrial dynamics in the epigenetic regulation and survival of AML LSCs.

Regardless of mitochondrial dynamics, it has been shown that AML LSCs have low levels of ROS [73]. This finding is supported by another study, which revealed that LSCs exhibit elevated levels of the enzyme glutathione peroxidase, known for its role as a ROS scavenger [81]. Adane and colleagues reported the presence of NOX2, a ROS-generating enzyme that is localized to the cytoplasm, in both bulk AML cells and LSCs [82]. This finding suggests that the site of ROS generation—whether in the mitochondrial electron transport chain or through cytosolic enzymes—may play a role in the maintenance of AML LSCs.

LSCs serve as a reservoir for the production of AML myeloblasts, which are highly proliferative [83]. Given the differences in their proliferative rates and characteristics compared to myeloblasts, it is important to understand the characteristics and metabolic dependencies of HSCs, LSCs, and AML myeloblasts (Figure 1). These dependencies include energy production, mitochondrial turnover, and sensitivity to ROS. A better understanding of AML LSC metabolism could aid in the identification of drugs that selectively target LSCs without harming HSCs, potentially improving AML treatment outcomes.

## 3. Altered Metabolic Pathways in AML

AML cells undergo profound metabolic reprogramming, a hallmark of cancer that supports uncontrolled growth and survival. This reprogramming involves significant alterations in various metabolic pathways that sharply diverge from the normal metabolic processes observed in normal hematopoietic cells.

### 3.1. Glycolysis

Non-cancerous cells metabolize glucose to carbon dioxide by the oxidation of glycolytic pyruvate in the mitochondria in the presence of oxygen. However, these cells produce increased amounts of lactate in the absence of oxygen. On the other hand, cancer cells, being highly proliferative, produce lactate irrespective of the availability of oxygen. This phenomenon has been termed the Warburg effect, where cancer cells preferentially produce energy through glycolysis followed by lactic acid fermentation in the cytosol, even in the presence of oxygen [84,85]. Thus, the Warburg effect, also termed “aerobic glycolysis”, meets the high glucose demand to keep up with the high proliferating rates of cancer cells [86]. The bioenergetic shift from the Warburg effect provides the metabolic requirements required for rapid cell proliferation. Glycolysis begins with the import of glucose into the cells through glucose transporters. Glucose is then phosphorylated to glucose-6-phosphate by hexokinase, trapping the glucose molecule within the cell. A vast majority of the trapped glucose is converted to pyruvate, which converts to lactate by lactate dehydrogenase (LDH) [87]. However, a small portion of the trapped glucose is channeled to pathways critical for ribose synthesis, phosphoglycerol synthesis, protein glycosylation, serine, and glycine synthesis. The enhanced glycolysis is often associated with the over-expression of both glucose transporters and glycolytic enzymes [88].

AML cells exhibit a similar increased glycolytic activity contributing to the aggressive phenotype of the disease. The metabolic shift is linked to the upregulation of specific enzymes like hexokinase 2 and pyruvate kinase M2 (PKM2), which play pivotal roles in enhancing glycolysis and redirecting glucose-derived carbons towards anabolic processes [89,90,91]. Hexokinase 2 catalyzes the first step of glycolysis, converting glucose to glucose-6-phosphate, which is a rate-limiting step in the pathway. PKM2, on the other hand, catalyzes the final step, converting phosphoenolpyruvate to pyruvate, and its activity is pivotal in determining whether the cell proceeds with aerobic respiration or switches to lactic acid fermentation. The reduction in the coenzyme NAD+ to NADH generated during glycolysis further optimizes energy production from glycolysis [92,93]. Importantly, metabolic reprogramming of glycolysis in AML is often connected to oncogenic signaling pathways, such as those activated by FLT3-ITD mutations, which can upregulate glycolytic flux independently of oxygen availability [94]. These alterations not only fuel cell proliferation but also contribute to the suppression of apoptosis, creating a cellular environment conducive to leukemia progression.

The shift in energy metabolism in AML cells also contributes to the suppression of apoptosis. The overactivity of glycolytic pathways, particularly through the stabilization of hypoxia-inducible factor 1-alpha (HIF-1α) under normoxic conditions, promotes survival signals and inhibits cell death pathways, thereby contributing to leukemia progression and chemotherapy resistance [95,96]. Understanding the role of glycolysis in AML has significant therapeutic implications. Targeting metabolic pathways offers a promising strategy for cancer therapy, and inhibitors of glycolysis, such as 2-Deoxy-D-glucose (2-DG), have been proposed as potential treatments for AML. These approaches aim to disrupt the energy supply to leukemic cells, thereby reducing proliferation and inducing apoptosis. In Figure 2, we illustrate the glycolysis biosynthetic pathway and the existing glycolytic inhibitors. The reliance of AML cells on glycolysis and the perturbation of specific glycolytic enzymes presents a complex but novel and targetable aspect of leukemia biology.

### 3.2. The Tricarboxylic Acid (TCA) Cycle

In AML, the tricarboxylic acid (TCA) cycle, also known as the Krebs cycle, is crucial for cellular energy generation and is restructured to prioritize biosynthetic needs associated with rapid cell division and growth, diverging from its typical role in energy production. The primary role of the TCA cycle is to generate energy in the form of ATP, as well as to provide precursors for various biosynthetic pathways. In normal hematopoietic cells, the TCA cycle begins with the synthesis of citrate, formed by the condensation of oxaloacetate (OAA) with acetyl-CoA, which is efficiently generated from pyruvate in the mitochondria [97,98]. This cyclic reaction facilitates the transfer of high-energy electron carriers, such as NADH, to the mitochondria’s ETC, which is crucial for energy production. Essentially, this process establishes a direct connection between glycolysis, particularly through its end product pyruvate, and the TCA cycle [98]. Alternatively, AML cells can preferentially perform aerobic glycolysis, resulting in the conversion of pyruvate to lactate [99]. Initiation of the TCA cycle can be blocked by insufficient levels of pyruvate being converted to acetyl-CoA. This inadequacy arises when pyruvate generated from glycolysis is not efficiently transformed into acetyl-CoA. Consequently, in AML cells, glycolysis and the TCA cycle frequently become dissociated. As a result, AML cells often turn to other carbon-producing processes, such as glutamine metabolism and lipid metabolism, which provide diverse fuels for the TCA cycle [99].

The uptake of glutamine and glutaminolysis—the conversion of glutamine to glutamate and α-ketoglutarate (α-KG)—is a key process in AML cells that replenishes intermediates fed into the TCA cycle [100]. Studies have demonstrated that the accumulation of α-KG plays an important role in limiting leukemia progression. For instance, an enriched protein in AML cells called BCAA transaminase 1 (BCAT1) relocates α-amino groups from branched-chain amino acids to α-KG, effectively reducing α-KG levels [101]. The lentiviral knockdown of BCAT1 in leukemia stem cells (LSC) unsurprisingly led to an accumulation of α-KG, impairing LSC survival. AML LSCs with BCAT1 knockdown also demonstrated poor leukemia-initiating potential when transplanted into NSG mice. Further analysis demonstrated that BCAT1 stabilizes HIFα, which is essential for maintaining LSCs. Accordingly, BCAT1 knockdown induces HIFα protein degradation, resulting in decreased leukemia-progression ability [101]. By contrast, overexpression of BCAT1 decreased intracellular levels of α-KG in AML cell lines. The prolonged overexpression of BCAT1 for 10 to 20 weeks resulted in DNA hypermethylation due to diminished TET2 activity caused by the limited availability of intracellular α-KG. Prognostic effects of high BCAT1 levels were only observed in mice harboring primary AML cells without preexisting isocitrate dehydrogenase (IDH) or TET2 mutations. Understanding the relationships among BCAT1, αKG levels, TET2 activity, and DNA methylation in AML cells may identify strategies to reduce AML stem cell function.

Similarly, mutations in genes encoding for enzymes of the TCA cycle, particularly IDH 1 and IDH2, are notable for their role in the metabolic shift of AML cells. These mutations result in the aberrant production of oncometabolite 2-hydroxyglutarate (2-HG), which has far-reaching implications in cellular metabolism and epigenetics [102,103]. 2-HG acts as a competitive inhibitor, interfering with the normal function of DNA and histone demethylases, thus promoting hypermethylation and leading to altered gene expression patterns that can initiate and sustain leukemogenic processes [104]. Overall, the reprogramming of the TCA cycle in AML supports the anabolic processes necessary for rapid cell division and tumor growth. The shift from catabolism to anabolism in AML cells involves increased consumption of intermediates for macromolecule biosynthesis. In Figure 3, we illustrate the cellular energy metabolic pathway, particularly the TCA cycle and some of the emerging inhibitors targeting metabolism in AML.

### 3.3. Amino Acid Metabolism

Amino acids are essential substrates of biosynthetic reactions and necessary building blocks required for key pathways in healthy cells, such as cell growth, immunity, and metabolism. Due to the rapid growth and metabolism of cancer cells, they often exhibit increased demand for amino acids. Thus, amino acid restriction is an avenue with high therapeutic potential. For AML cells specifically, research has focused on dysregulated utilization of glutamine, arginine, asparagine, and tryptophan.

Current research continues to explore the role of amino acid metabolism in AML, with a focus on integrating metabolic therapies with existing chemotherapeutic regimens. The combination of metabolic inhibitors with chemotherapy, targeted therapies, or immunotherapy may enhance treatment efficacy and overcome resistance, paving the way for more effective and sustainable AML treatment strategies. The metabolism of amino acids, particularly glutamine, in AML cells represents a vital aspect of the cellular metabolism that supports rapid cell proliferation and survival under continuous stress conditions. Targeting these metabolic pathways provides a promising avenue for the development of novel therapeutic strategies in AML.

#### 3.3.1. Glutamine

Glutamine is most frequently associated with the proliferation and survival of AML cells [105]. Compared to non-cancerous cells, AML cells consume glutamine at a significantly greater rate to fuel the TCA cycle, synthesize nucleotides, and maintain the redox balance within leukemic cells, among other functions [100,105]. Glutamine is involved in the biosynthesis of glutathione and other amino acids [106,107]. Glutamine is produced de novo and also through the lysosomal degradation of proteins produced from autophagy, endocytosis, or micropinocytosis. Glutamine can also be imported by glutamine importer solute carrier family 1, member 5 (SLC1A5). SLC1A5 can be upregulated by lactic acid through the stabilization of HIF-2α to increase glutaminolysis. Cancer cells depend on solute carrier superfamily transporters located on cell membranes for the absorption of glutamine from the extracellular environment [108]. Leukemic cells utilize glutamine anaplerosis through glutaminolysis to glutamate in order to meet the increased metabolic needs [105]. Anaplerosis is a metabolic process that replenishes intermediates of a metabolic pathway, particularly the TCA cycle, which can be depleted for biosynthetic reactions. Glutamine enters the mitochondria and is degraded to glutamate through MYC-driven glutaminase (GLS). GLS has been shown to be required for lymphoma cell proliferation [109]. Glutamate undergoes oxidative deamination to produce α-KG and ammonia by the enzymes glutamate dehydrogenase (GDH) and transaminase. The α-KG thus generated enters the TCA cycle to produce malic acid, which is transported to the cytoplasm. Malic acid can further be converted to aspartate or citric acid and provides the material basis for energy through aerobic glycolysis [110].

Once glutamine is converted into glutamate by the enzyme GLS, glutamate is able to serve multiple functions in AML cells, such as (1) directly converting to α-KG, (2) serving as a nitrogen source following deamination, and (3) regulating the signaling events involved in mTORC1 activity. In addition to replenishing the TCA cycle, α-KG derived from glutaminolysis can serve as a carbon donor in various pathways frequently utilized by cancer cells, including (1) fatty acid synthesis, (2) the reduction in NADP+ to NADPH, and (3) the production of glutathione, a protective agent against ROS. In addition to generating α-KG to support ATP production through OXPHOS, glutamine plays an important role in anabolism [111,112].

Several studies have provided direct evidence suggesting the importance of glutamine metabolism for AML progression. As previously discussed, the knockout of SLC1A5, a key glutamine importer, led to the apoptosis of AML cell lines and restricted tumor development in AML xenografts [113]. Stromal cells acquire carbon and nitrogen from noncanonical sources to synthesize glutamine. Another study focused on the effects of a CB-839, a GLS inhibitor. CB-839 was found to inhibit glutathione production, leading to the accumulation of mitochondrial ROS and apoptotic cell death [114]. Interestingly, CB-839 synergized, in vitro and in vivo, with chemotherapeutic agents that further induce oxidative stress, such as arsenic trioxide and homoharringtonine [114]. Evolving knowledge about glutamine metabolism suggests that targeting this metabolic pathway may limit leukemia progression.

#### 3.3.2. Asparagine

Although glutamine is currently the most widely studied amino acid in AML metabolism, a few studies have focused on the importance of other dysregulated amino acids, namely arginine and asparagine. During normal physiology, arginine plays a role in cell division, wound healing, immune function, and the biosynthetic pathways of different hormones. Additionally, it serves as a polyamine precursor. A majority of AML cells do not produce arginine de novo due to the absence of argininosuccinate synthetase-1 (ASS1), which is required to synthesize arginine from citrulline [115,116]. The de novo arginine produced, if any, is not sufficient to meet the needs of proliferating AML cells. This condition makes the cancer cells partially auxotrophic by arginine-depleting agents [117]. Thus, AML blasts depend on exogenous arginine sources and, thus, constitutively express cationic amino acid transporters (CAT), specifically CAT1 and CAT2, for arginine import [116]. Arginine is metabolized through the urea cycle into polyamines, which are essential components of cell growth and proliferation. Mitochondrial Arg2 catalyzes the hydrolysis of arginine and this enzyme has been found to be overexpressed AML blasts [118]. While the dependence on arginine metabolism by AML blasts has not been extensively investigated, several key studies have been carried out. Mussai et al. demonstrated that AML blasts maintain an immunosuppressive microenvironment through enhanced arginine metabolism [119]. This study found that AML blasts secrete high levels of arginase II, which catalyzes the conversion in the urea cycle of L-arginine to urea and L-ornithine. Arginase II was found to suppress T-cell proliferation and polarize monocytes to an immunosuppressive M2-like phenotype both in vitro and in vivo. Moreover, this study found increased plasma arginase II levels in AML patients [119]. Therapy that is able to target the immunosuppressive ability of AML cells could lead to improved outcomes for AML patients. Overall, though, arginine metabolism dependency in AML is still in need of further investigation.

Asparagine is synthesized by the transamidation of aspartic acid by the enzyme asparagine synthase (ASNS). The downregulation of the gene encoding for ASNS due to the abnormal monosomy of chromosome 7 leads to reduced biosynthesis of L-asparagine from L-aspartate. Neoplastic cells, however, require high levels of asparagine to maintain high proliferative growth. Specifically, intracellular asparagine levels have been found to be critical for cancer cell growth due to its role as an amino acid exchange factor required for the antiport of multiple essential amino acids such as glutamine, arginine, and serine. However, some neoplastic cells actually lack ASNS and, thus, require exogenous sources of asparagine. For instance, acute lymphoblastic leukemia (ALL) cells tend to exhibit low levels of ASNS and are usually very sensitive to asparagine depletion. Thus, asparagine-depleting agents are of interest in potentially limiting AML cell proliferation.

#### 3.3.3. Tryptophan

Tryptophan is an essential amino acid and, therefore, needs to be obtained from consuming food sources containing it. In the cells, tryptophan is broken down to kynurenine, with the rate-limiting enzymes being indoleamine-2,3-dioxygenase (IDO) 1, IDO2, and tryptophan-2,3-dioxygenase [120]. In cancer cells, there is an upregulation of IDO and tryptophan-2,3-dioxygenase, which promotes tumor growth and immune-suppressive environment [121]. Because of the role played by IDO in suppressing the tumor microenvironment, inhibitors blocking them are being used in AML [122]. Higher expression of IDO is associated with significantly decreased overall and relapse-free survival [123]. IDO inhibitors, such as 1-methyl tryptophan (1MT), aid in mounting an effective immune response by reversing the immunosuppressive functions of IDO, resulting in the transformation of T cells into T reg cells [124]. IDO inhibitors possibly have other mechanisms of action apart from modulating the tumor microenvironment, and this area needs further evaluation.

### 3.4. Lipid Metabolism

#### 3.4.1. Fatty Acid Biosynthesis and Oxidation

Cellular and sub-cellular membranes primarily consist of cholesterol, phospholipids, and triglycerides. These lipid components of cell membranes can be sourced from the blood in the form of free fatty acids and lipoproteins. However, cancer cells have a heightened need for the production of structural membrane components, which requires active lipid biosynthesis. The first step in the fatty acid synthesis pathway involves the carboxylation of acetyl-CoA, which converts it to malonyl-CoA [125]. Malonyl-CoA then undergoes a condensation reaction to form palmitate, facilitated by the enzyme fatty acid synthase (FASN). The fatty acid biosynthetic pathway produces essential fatty acids required for the generation of triglycerides, glycerophospholipids, cardiolipins, sphingolipids, and eicosanoids.

Cancer cells not only rely on fatty acid biosynthesis but also on fatty acid oxidation (FAO), which is the process of breaking down fatty acids. This process primarily occurs in the mitochondria, where fatty acids are degraded to produce acetyl-CoA. Acetyl-CoA plays a crucial role in energy production and OXPHOS. Biomass and energy produced from fatty acid (FA) metabolism are heavily utilized by AML cells to meet the demands of their rapid proliferation. Both FA metabolism and FAO are dysregulated pathways in AML cells that promote leukemogenesis. AML cells rely on FAO as the β-oxidation of FAs produces acetyl-CoA and yields significant levels of ATP compared to glucose oxidation. The growth and survival of acute monocytic leukemia cells may be supported by bone marrow adipocytes, which promote fatty acid β-oxidation [126]. Prolyl Hydroxylase 3 (PHD3), an α-KG-dependent dioxygenase, activates acetyl CoA carboxylase 2 (ACC2), leading to decreased FAO [127]. In AML cells, PHD3 expression is frequently reduced, which supports increased FAO. German et al. demonstrated that PHD3 overexpression leads to reduced FAO and subsequently decreased AML proliferation [127]. Compared to quiescent hematopoietic stem cells, AML cells exhibit high FAO rates and target FAO pharmacologically sensitized human leukemia cells to apoptosis induction [128]. Samudio and colleagues showed that inhibiting FAO with etomoxir or ranolazine suppressed the proliferation of leukemic cells and also made them more sensitive to ABT-737, a molecule that triggers the release of pro-apoptotic Bcl-2 proteins like Bak [128]. Additionally, treatment of leukemia cells with the fatty acid synthase or lipolysis inhibitor orlistat also sensitized them to ABT-737. These data suggest that fatty acids promote cell survival and that FAO inhibitors are potential therapeutic strategies for hematological malignancies such as AML.

Additionally, FAO may enhance chemotherapy resistance in AML cells. In mice harboring AML from patient-derived xenografts, the metabolic profile of cells resistant to cytarabine revealed upregulated FAO and high CD36 expression [35]. CD36 is a key FA transporter that facilitates the uptake of FAs into the cell. FAO has also been shown to be important in the mechanism of Aza/Ven resistance, which is frequently seen in relapsed AML patients. LSCs from de novo vs. relapsed patients were examined in culture without amino acids. LSCs from relapsed AML patients demonstrated high levels of FAO to compensate for amino acid restriction and had a resultant reduction in OXPHOS, while de novo AML LSCs did not demonstrate this shift. Importantly, there was only a minor reduction in the viability of LSCs treated with Aza/Ven in the relapsed patients. This was attributed to these increased FAO resistance mechanisms [129].

Importantly, in the FAO pathway, certain enzymes, such as carnitine palmitoyl transferase 1a (CPT1a) and the carnitine transporter (CPT2), play crucial roles. The process of transferring fatty acids to the mitochondria involves coupling them with coenzyme A and subsequently transferring the acyl group to carnitine. CPT1a is responsible for catalyzing this reaction. It was shown that CPT1a is highly expressed in AML cells compared to non-cancerous cells. Riccardi and colleagues demonstrated that the CPT1a inhibitor, ST1326, induces cell growth arrest, mitochondrial damage, and apoptosis in AML cells [130]. Therefore, targeting fatty acid transport and β-oxidation has been identified as an additional potential therapeutic strategy for AML.

#### 3.4.2. Lipid Steroids

Lipid steroids (LS) play crucial roles in various biological processes within cells. Cholesterol, a type of lipid steroid, contributes to membrane structure and helps maintain membrane fluidity. Certain lipid steroids are integral components of the electron transport chain, such as ubiquinone. Additionally, they are involved in protein glycosylation, while others, like farnesyl pyrophosphate and geranylgeranyl pyrophosphate, assist in protein isoprenylation. Lipid steroids are synthesized in the mevalonate pathway from acetyl-CoA. Pharmacologically, statins inhibit lipid steroid synthesis by blocking the enzyme HMG-CoA reductase (HMGCR), which converts HMG-CoA to mevalonate [131].

Moreover, statins that target LS have been shown to induce apoptosis in hematological malignancies [132]. Lovastatin has been shown to induce a differentiation response in AML cells [133]. Further research demonstrated that lovastatin induces apoptosis in AML cells by inhibiting protein geranylgeranylation [134]. Additionally, Wong and colleagues revealed that statins act as inhibitors of HMGCR, leading to tumor-specific apoptosis [135]. In addition to action related to inhibition of HMGCR, statins have been shown to activate the mitochondrial pathway in human lymphoma blasts and myeloma cells by interfering with the prenylation of Rho and Ras GTPases [136].

Beyond their use as a single agent, the effectiveness of statins in combination with chemotherapy has also been evaluated. The SWOG S0919 study was a phase 2 trial examining the combination of idarubicin and cytarabine with pravastatin for the treatment of AML [137]. In this trial, high-dose statin combined with idarubicin resulted in a 75% complete remission rate in relapsed patients with favorable risk profiles. However, in those categorized as poor risk due to adverse cytogenetics and molecular mutations, the response rate was only 30%. These findings suggest that inhibiting cholesterol synthesis could offer therapeutic benefits in the treatment of AML [137].

#### 3.4.3. Sphingolipids

Bioactive sphingolipids offer a promising new approach for treating AML. Sphingolipids are an essential component of cell membranes, playing a crucial role in controlling cell proliferation, facilitating cell–cell interactions, and regulating signaling and cell survival. In the sphingolipid biosynthetic pathway, de novo synthesis begins with the combination of palmitoyl-CoA and serine, resulting in the formation of sphingosine, which serves as the backbone of sphingolipids. This reaction is catalyzed by the enzyme serine palmitoyltransferase (SPT). Sphingosine then conjugates with a fatty acid to form ceramide via enzymes acid ceramidase (AC) and ceramide synthase (CerS). The synthesis of ceramides—sphingolipids that are important for signal transduction and apoptosis—is catalyzed by AC. Research by Obeid et al. has highlighted the pro-apoptotic potential of ceramides and their roles in promoting cell cycle arrest and inducing cellular senescence [138]. Elevated levels of AC have been found to enhance the survival of AML cells by increasing the activity of the anti-apoptotic protein MCL-1. As a result, treatment with the AC inhibitor LCL204 significantly reduced AML cell viability and decreased disease burden in mice harboring primary AML cells [139].

In addition to the AC enzyme, the CerS enzyme has also been shown to be important, especially in FLT3 ITD AML. FLT3 signaling is linked to decreased CerS expression, which can be restored by inhibiting FLT3 [140]. Furthermore, FLT3-ITD mutations in AML cells are associated with the inhibition of pro-apoptotic ceramides, which are vital for the execution of cell death processes. Consequently, targeting FLT3 mutations could be an effective strategy to reactivate ceramide pathways and promote apoptosis in AML cells [140]. This approach may help to overcome the chemotherapy resistance seen in primary AML cells with FLT3 mutations. Sphingosine-1-phosphate (S1P) is a secreted sphingolipid that plays a vital role in cell migration and the tissue homing of lymphoid and myeloid cells. S1P is produced when sphingosine is phosphorylated by sphingosine kinase (SPHK). Inhibition of SPHK, combined with the chemotherapy drug cytarabine, has been shown to induce the death of AML cells, highlighting the importance of S1P signaling in leukemia [141,142]. Overall, these findings demonstrate that lipid metabolism is a dependency of AML progenitor cells. Thus, targeting these pathways is worthy of further pre-clinical evaluation.

## 4. Current and Emergent Metabolic Therapeutic Strategies

Previous studies have started to delineate the complex metabolic processes that AML cells undertake to proliferate at exorbitant rates. Therapeutics targeting the altered metabolic pathways of AML cells present a novel approach to improving AML outcomes. Extensive research has already pinpointed and advanced several therapeutics that exploit the metabolic vulnerabilities of adult AML cells. Such advances have targeted glycolysis, amino acid metabolism, and lipid metabolism and have been relatively effective in inhibiting leukemia progression. However, the translation of these strategies into clinical practice faces several challenges, reflecting the complexity of AML’s metabolic network and the heterogeneity of the disease. Thus, increased pre-clinical studies should continue to explore AML metabolic plasticity to further improve outcomes for this disease.

### 4.1. Agents Targeting Carbohydrate, Amino Acid Metabolism, and Fatty Acid Metabolism

Several therapeutic agents that target various substrates involved in the altered metabolic pathways of AML cells have been developed. These drugs aim to impair AML energy production while keeping healthy hematopoietic cells relatively undisturbed. As described, AML cells rely on high levels of glucose metabolism, and this reliance is well understood. Therefore, therapeutic strategies that target glycolytic enzymes to inhibit glycolysis and ultimately induce apoptosis in leukemic cells are promising. Hexokinase 2, a rate-limiting enzyme of glycolysis, performs the first committed step in the glycolysis pathway, and its kinetic properties help determine the pace at which glucose is metabolized by cells. Inhibition of hexokinase 2 by 2-DG decreased AML cell proliferation and increased AML sensitivity to cytarabine in both AML primary blasts and AML cell lines [143,144]. 3-bromopyruvate (3BrPA) also inhibits hexokinase 2 and has been extensively studied on Adriamycin (ADR)-resistant AML cell lines with high glycolytic activity. By reducing intracellular ATP levels through the inhibition of glycolytic activity, 3BrPA rendered ADR-resistant AML cells more susceptible to the cytotoxic effects of ADR [145]. Given that Adriamycin, also known as doxorubicin, is an anthracycline chemotherapy agent, these findings may be applicable to other anthracycline-resistant AML cells. Although extensive research has focused on inhibiting glycolysis in newly diagnosed AML patients, further studies are necessary to understand its effects on relapsed patients and LSCs.

Significant uptake of glutamine is a key metabolic characteristic of AML cells. Agents that can inhibit energy production via glutaminolysis represent another area of interest. Telaglenastat (CB-839) is an oral glutaminase inhibitor that was found to inhibit both GLS1, an important enzyme in increasing intracellular glutamine levels, and glutathione, an important antioxidant that neutralizes the high levels of ROS in AML cells. CB-839 led to decreased survival in AML cell lines, primary AML patient samples, and AML xenografted mice [114]. A recent phase I clinical trial demonstrated tolerability of CB-839 in AML patients [146]. Additional clinical trials of CB-839 as a single agent and in combination with cytotoxic chemotherapies are ongoing. Interestingly, glutaminase inhibition is synergistic with several chemotherapeutic drugs, including arsenic trioxide and venetoclax, suggesting potential novel therapeutic combinations [114,115]. Understanding and targeting glutamine metabolism may have important therapeutic effects for limiting leukemia progression.

Agents targeting arginine and asparagine are currently under investigation or are part of currently utilized treatment plans. The reduction in arginine with arginine deaminases ADI-PEG 20 and BCT-100 demonstrated anti-leukemic effects both in vivo and in vitro [115,116]. Importantly, BCT-100 in combination with cytarabine exhibited greater cytotoxicity than the sum of the individual compounds alone in primary blasts from patients [116]. L-asparaginase, which converts asparagine to aspartic acid, has anti-leukemic effects and has been used as a part of leukemia treatment plans for decades for pediatric AML [147,148]. One clinical trial combining L-asparaginase with high-dose cytarabine and mitoxantrone has resulted in positive outcomes in elderly AML patients [147]. Altogether, these studies underscore the importance of amino acid metabolism in AML and may serve as promising avenues for novel treatment options.

Lastly, agents that suppress FAO and FAS have been identified and developed and have demonstrated anti-leukemic effects in the pre-clinical setting. Carnitine palmitoyl transferase 1A (CPT1A) is a rate-limiting enzyme in FAO that facilitates the transport of fatty acids into the inner membrane of the mitochondria. The overexpression of CPT1A is associated with adverse outcomes in AML patients, making it an attractive target for drug development [149]. Three CPT1A-inhibitory agents—avocatin B, etomoxir, and ST-1326—have been shown to restrict leukemic proliferation in AML pre-clinical models. Avocatin B—a lipid derived from avocados—has recently been identified as a novel anti-leukemia compound. Avocatin B preferentially targets AML cells over normal cells due to the differing mitochondrial characteristics of these cells [150]. This agent specifically inhibits fatty acid oxidation, leading to reduced levels of NADPH and GSH, which are crucial for counteracting ROS. The decrease in these antioxidants results in an increase in ROS, which induces apoptosis in leukemia cells [150]. Another agent targeting lipid metabolism is etomoxir [128], a protein that inhibits CPT1A and thus blocks FAO. Etomoxir has been shown to decrease the viability of AML cell lines by preventing ATP production through FAO [151]. Lastly, the drug ST-1326 has slowed the proliferation of leukemic cells, which results in their apoptosis, sensitizing these malignant cells to the cytotoxic effect of cytarabine. In Table 1, we listed the emerging compounds targeting metabolic pathways in AML, their mechanism of action and stage of development. Overall, when compared to normal HSCs, leukemia cells have a higher mitochondrial mass and a greater dependence on fatty acids, which makes them more susceptible to treatments that target mitochondrial fatty acid oxidation [71].

### 4.2. OXPHOS Inhibitors: Mechanisms and Clinical Implications

As studies demonstrate the reliance on high OXPHOS in LSCs and highly proliferative AML cells, agents that inhibit OXPHOS are becoming of great interest. BCL-2 is an anti-apoptotic protein, and its inhibition has been shown to impede OXPHOS [73]. Thus, agents that are able to preferentially obstruct BCL-2 function may target LSCs in AML patients. An extensively studied and FDA-approved BCL-2 inhibitor is venetoclax [164]. Venetoclax selectively binds to BCL-2 with high avidity, blocking its normal function. It has been tested in various clinical settings, including a phase III study where venetoclax combined with azacitidine significantly improved survival and remission rates in adult AML patients who had never received chemotherapy compared to those receiving azacitidine alone [156]. Furthermore, in other recent studies for de novo AML patients, the combination of venetoclax with intensified treatment regimens like daunorubicin and cytarabine (DA) or fludarabine, cytarabine, granulocyte-colony stimulating factors, and idarubicin (FLAG-IDA) has shown remarkable efficacy. For instance, a regimen combining venetoclax with DA achieved a combined complete remission and complete remission with an incomplete hematologic recovery rate of 91% across 33 patients, with a one-year overall survival rate of 97% [165]. Similarly, the venetoclax + FLAG-IDA treatment in 45 patients demonstrated a 98% overall response rate [166].

Another agent that has undergone clinical evaluation is IACS-010759. This drug inhibits ETC1 activity by binding at the entrance of the ubiquinone channel, thereby blocking ubiquinone function. In both AML and other cancer cell lines, treatment with IACS-010759 resulted in reduced cancer cell growth and a modest increase in apoptosis. The compound was well-tolerated and enhanced survival in AML mouse models. Pre-clinical testing of IACS-010759 in mice, rats, dogs, and cynomolgus monkeys demonstrated a tolerable toxicity profile. These data established the human starting dose according to FDA guidelines [167], and a phase I study of IACS-010759 for relapsed AML patients and patients with advanced solid tumors was completed [152]. Unfortunately, this study identified significant toxicity, especially those related to elevated blood lactate and neuropathy, associated with achieving target IACS-010759 plasma exposures. Subsequent investigations revealed that effective doses of IACS-010759, along with its coadministration with an HDAC6 inhibitor, induced peripheral neuropathy in mouse models [152]. Consequently, the development and further study of other potential OXPHOS inhibitors with better side effect profiles are imperative.

A new therapeutic option of interest is the OXPHOS inhibitor atovoquone. Atovoquone (AQ) is an FDA-approved anti-microbial drug that is an indication for the treatment of Pneumocystis jiroveci pneumonia (PJP). Adults with AML receiving AQ for PJP prophylaxis had lower relapse rates of their AML [168]. AQ suppresses oxidative phosphorylation, induces apoptosis in pediatric AML cells, and prolongs survival in pediatric AML xenografts [169]. AQ is well tolerated and does not cause hepatotoxicity, nephrotoxicity, or myelosuppression. However, enteral absorption issues driven by anorexia, impaired mucosal membrane health, and resistance to consumption of a fatty diet while on intensive therapy may hinder the potential clinical benefits of incorporating AQ into upfront intensive pediatric AML therapy. Based on this, a clinical trial was conducted (NCT03568994) that evaluated the feasibility of incorporating AQ with standard upfront pediatric AML chemotherapy. This trial found that achieved plasma AQ levels were lower than target levels of 40–80 µM when administered with concomitant cytotoxic chemotherapy. Nonetheless, administration of AQ was demonstrated to be feasible and safe in pediatric AML patients. We illustrated the currently used OXPHOS inhibitors in Figure 4.

### 4.3. Agents Targeting Genomic Aberrations Associated with Metabolic Pathways

AML is a highly heterogeneous disease with a variety of genetic mutations that significantly impact metabolic pathways. These mutations not only affect the proliferative capabilities of leukemic cells but also their survival strategies, altering fundamental metabolic processes such as the TCA cycle. The impact of genetic mutations on the metabolic pathways of AML highlights the complexity and diversity of this malignancy. Understanding how these mutations alter cellular metabolism provides critical insights into the disease’s biology and opens up new avenues for targeted therapies.

One of the most studied mutations in AML involves the isocitrate dehydrogenase (IDH) genes, specifically IDH1 and IDH2 [170]. IDH exists in three isoforms: IDH1, IDH2, and IDH3. So far, IDH3 has not been shown to be associated with malignancies. It plays a role in the Krebs cycle, converting NAD+ to NADH in the mitochondria and consists of heterogeneous subunits. In contrast, the isoforms IDH1 and IDH2 perform their reactions outside the Krebs cycle, using NADP+ as a cofactor. These isoforms form homodimers and are known to be associated with AML. IDH catalyzes the oxidative decarboxylation of isocitrate to produce α-ketoglutarate (α-KG) and carbon dioxide (CO_2_) [104]. This process involves the oxidation of isocitrate to oxalosuccinate, followed by the decarboxylation of the carboxyl group to form α-KG. IDH2 mutations are typically heterozygous and cause a change in substrate binding. As a result, the enzyme can no longer bind to isocitrate; instead, it binds to α-KG and converts it into 2-HG [171]. The accumulation of 2-HG has profound implications, as it can inhibit a family of dioxygenases, including histone lysine demethylases and the TET family of DNA hydroxylases, leading to epigenetic changes that promote leukemogenesis [172]. 2-HG also acts as a competitive inhibitor of α-KG-dependent dioxygenases, thereby affecting cell differentiation and promoting the survival of leukemic cells [173]. IDH1 mutations have been shown to accelerate the onset of myeloid leukemia-like disease in mice, and they interact with HoxA9, resulting in accelerated cell cycle progression [174]. Similarly, IDH2 mutations also cooperate with HoxA9 and Meis1a, leading to the initiation and maintenance of leukemia [175]. Furthermore, mutations in IDH have been associated with a poorer prognosis in patients with cytogenetically normal AML who have NPM1 mutations but do not have FLT3 tandem duplications [176]. IDH2 mutations produce mitochondrial NADPH from NADP+, which plays a key role in the recycling of glutathione. Glutathione levels are known to regulate the redox balance in cells. The recognition of IDH1/2 mutations as central to the metabolic reprogramming of AML cells has led to the development of targeted inhibitors.

Mutated IDH 1 and 2 have been found in ~20% of AMLs and thus constitute another therapeutic target of interest [118]. Inhibitors of IDH 1/2 mutations have demonstrated promising results in several clinical trials. Several clinical trials featuring agents that target IDH1 mutations, such as Ivosidenib (AG-120), BAY-1436032, and Olutasidenib (FT-2102), have determined the efficacy of these drugs in combination strategies. Ivosidenib in combination with Azacitidine in a phase III trial improved event-free survival, response, and overall survival compared with placebo and azacitidine-only treatment in mutant-IDH1 AML patients (NCT03173248) [177]. AG-120 demonstrated similar response rates and rates of differentiation to those of patients receiving AG-221 [36]. Similarly, FT-2102 is an FDA-approved oral small-molecule inhibitor of IDH1 for treating adult patients with relapsed or refractory AML. FT-2102 demonstrated an improved median survival rate and complete remission plus complete remission with partial hematologic recovery rates compared to similar drugs, like AG-120. However, FT-2102 carries notable risks, including grade 3–4 cytopenias, differentiation syndrome, liver enzyme elevations, and treatment-emergent adverse events leading to dose modifications or discontinuation in a significant proportion of patients [161]

Enasidenib (AG-221) is an orally bioavailable and potent inhibitor of the R140Q- and R172K-mutated IDH2 enzyme. In a pre-clinical study, AG-221 inhibited mutated IDH2, decreased 2HG production, and induced myeloid differentiation in AML cell lines and xenograft mouse models [178]. This agent has completed phase III clinical trial testing (NCT02577406) and exhibited positive outcomes for mutant-IDH2 relapsed and refractory AML patients receiving AG-221. Compared to conventional salvage therapies, AG-221-treated patients had improved 1-year survival, event-free survival, and morphological and hematological response [179]. AG-221 has been found to be well tolerated; however, ~14% of patients experienced differentiation syndrome, similar to the 12% of hematological malignancies found in the phase I/II study that established the dosing [23]. Agents targeting mutations in IDH1 mutations, such as AG-120 and FT-2102, and IDH2, such as AG-221 have shown promise in clinical trials, effectively lowering 2-HG levels, inducing cellular differentiation, and demonstrating clinical efficacy in patients with relapsed or refractory AML [23].

Beyond IDH1/2, other genetic aberrations also impact metabolic pathways in AML. The PI3K/AKT/mTOR signaling pathway is central to cellular growth and metabolism, playing a pivotal role in glucose uptake, glycolysis, and lipid synthesis. Targeting the mTOR pathway with rapamycin and its analogs has shown efficacy in preclinical AML models by reducing leukemic cell proliferation and inducing apoptosis. These effects are partly due to the inhibition of mTORC1, which leads to decreased protein synthesis and cell growth [180] Mutations in FLT3, TP53, and RAS influence various aspects of cellular metabolism, including glycolysis and mitochondrial function. For instance, FLT3-ITD mutations are associated with increased glycolytic flux, which supports rapid cell proliferation and survival under stress conditions [144,181]. These studies highlight the heterogeneity of the disease and demonstrate the effectiveness of drugs in a subtype-specific manner. Given the hematological toxicity associated with AML treatments, combinations involving antimetabolites and targeted therapies should be thoroughly investigated.

## 5. Challenges in Targeting Metabolism in AML and Clinical Translation

Although recent years have seen an upsurge in AML metabolic profiling, effectively designing metabolic therapies across diverse AML populations remains a challenge for several reasons. First, the heterogeneity of AML, both genetically and metabolically, complicates the development of universal metabolic therapies [182,183]. This necessitates the identification of biomarkers to select patients most likely to benefit from specific metabolic therapies [73]. Secondly, metabolic inhibitors can affect both cancerous and normal cells, leading to systemic toxicity. Selectively targeting AML cells while minimizing off-target effects is thus a potential challenge that must be addressed and carefully examined in rigorous pre-clinical and early-phase clinical testing [184]. A third challenge is that AML cells can adapt to metabolic inhibitors by activating compensatory pathways, leading to resistance. Thus, understanding these resistance mechanisms is critical for developing effective combination therapies [185]. In fact, the metabolic dependencies of AML cells have been shown to shift between de novo and relapsed disease. Thus, it is important that pre-clinical testing is conducted in both de novo and relapsed patient samples to provide rationale for which patient populations to include in clinical trials and to increase the chances of observing early efficacy signals. For all these reasons, while metabolically targeted therapies are very promising in AML, clinical trial design for these agents must carefully address a multitude of potential issues, including but not limited to optimal dosing, combination treatment regimens, and patient selection criteria. These factors are crucial for translating promising pre-clinical findings into clinical success [186].

Anti-metabolites as single agents have often demonstrated promising preclinical efficacy but failed to achieve similar success in clinical settings, largely due to significant off-target toxicity in healthy tissues [187]. For example, the OXPHOS inhibitor IACS-010759 induced severe adverse effects such as elevated blood lactate levels and neurotoxicity during clinical evaluation, underscoring the challenge of systemic metabolic inhibition [152]. These outcomes may stem from the essential role of metabolic pathways in normal cell function, making it difficult to selectively target leukemic cells without affecting healthy counterparts. One potential solution involves the use of antibody–drug conjugates or nanoparticle-based delivery systems that can improve malignant cell specificity by exploiting AML-specific surface markers [188]. In addition, anti-metabolic therapies can also affect immune cell function. Naïve T cells are heavily reliant on OXPHOS, whereas activated effector T cells shift toward glycolysis for rapid proliferation and function [189,190]. As such, metabolic drugs may inadvertently impair anti-tumor immunity. Furthermore, many early studies have focused on pathway-level inhibition (e.g., glycolysis or glutaminolysis) without a deep understanding of the downstream cellular responses they trigger. A more holistic approach—considering immune effects, off-target toxicity, and cell death mechanisms—will be critical for designing selective, effective metabolic therapies for AML.

The vulnerable subpopulation of pediatric AML (pAML) presents an especially unique challenge due to the distinct biology of the disease and the need for treatments that spare developing tissues [191,192]. Because pAML is driven primarily by cytogenic fusions and less by single gene mutations, therapies that are effective in adult AML do not always translate to pAML. This may hold true for metabolically targeted therapeutics as well [38]. Furthermore, the standard of care chemotherapy for pediatric AML is very intensive, with treatment-related mortality at 5–10%, so incorporating additional therapeutics into upfront or traditional salvage chemotherapy regimens has the potential to cross tolerability boundaries. This must be considered in the trial design for pAML. Logistically, drug delivery routes for pAML patients can also be an issue. Pediatric patients receiving enteral medications frequently require drug formulations that can be administered as a liquid, either in a suspension formation or with a tablet that can be crushed. Given that the age distribution of pediatric AML is bimodal with approximately half of pAML cases under the age of 3 years, this is an important consideration to ensure accessibility to promising agents for this patient population.

The unique metabolic characteristics of AML cells provide valuable targets for therapeutic intervention. While significant progress has been made in understanding and targeting these pathways, several challenges remain in translating these strategies into clinical practice. Addressing the heterogeneity of AML, overcoming resistance mechanisms, and ensuring selective targeting of leukemic cells are essential for the successful development of metabolic therapies. In particular, potential hematological toxicities—especially in the context of disease-induced bone marrow failure—must be carefully considered in future clinical studies. Furthermore, clinical experience with combining antimetabolic agents and conventional cytotoxic therapies remains limited, underscoring the need for systematic evaluation. The potential synergy between metabolism-targeting therapies and molecularly targeted agents also warrants further investigation. Identifying defined AML subsets that may benefit from these strategies will be critical. As ongoing research and clinical trials continue to refine these approaches, the rational development of selective, well-tolerated metabolic therapies holds significant promise for improving outcomes in patients with AML.

## 6. Conclusions

AML exhibits profound heterogeneity at both the genomic and metabolic levels, contributing to its complex therapeutic landscape. This heterogeneity includes significant inter- and intra-patient variability, which complicates the development of effective treatments. The presence of subclones within a tumor, each exhibiting distinct metabolic rewiring, further challenges our ability to target the disease effectively. Although significant strides have been made in identifying genomic abnormalities in AML, our understanding of how these mutations lead to altered metabolic processes remains incomplete. Gaining key insights into the energy metabolism of leukemic cells, including LSCs and therapy-resistant cells, is critical for advancing treatment strategies. There is an urgent need to unravel the complex metabolic networks within AML to design more effective and less toxic therapeutic approaches.

Cancer cells undergo genomic alterations that result in changes in transcription, protein expression, and protein function. The metabolome captures the functional readout of upstream changes in a cancer cell’s genome, transcriptome, and proteome. Altered metabolites can influence protein activity, which in turn affects RNA transcription and DNA replication. The application of advanced omics technologies—such as integrated metabolomics and proteomics—will be crucial for mapping these metabolic pathways in unprecedented detail.

Future studies should focus on implementing single-cell sequencing technologies to dissect the metabolic heterogeneity within AML subpopulations, including LSCs and resistant clones, to identify targetable metabolic vulnerabilities. Longitudinal studies tracking metabolic changes throughout treatment and at relapse are also needed to better understand the dynamics of metabolic rewiring and resistance mechanisms. Utilizing CRISPR-based functional genomics can systematically identify metabolic enzymes and pathways essential for AML survival and proliferation, offering new therapeutic targets. Exploring combination therapies that include metabolic inhibitors may help overcome the redundancy and plasticity of metabolic pathways, potentially preventing the development of resistance. Integrating metabolic therapies into clinical trials, with a focus on targeting specific metabolic pathways identified through genomic and metabolomics analyses, could help tailor treatments to individual metabolic profiles—enhancing efficacy and reducing adverse effects.

## Figures and Tables

**Figure 1 cancers-17-01355-f001:**
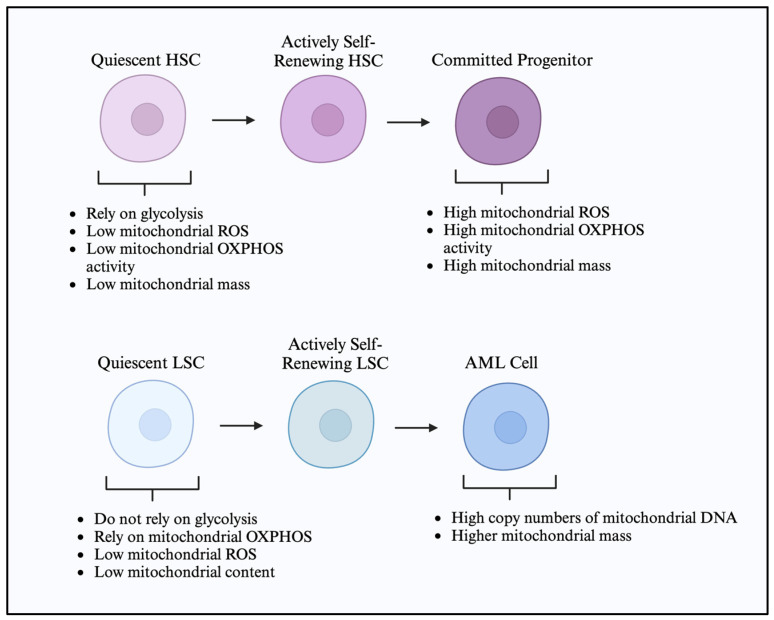
Metabolic differences between HSCs and LSCs. Metabolic differences between HSCs and LSCs are illustrated, highlighting their distinct metabolic activities during the quiescent and actively self-renewing states. Figure created with BioRender.com.

**Figure 2 cancers-17-01355-f002:**
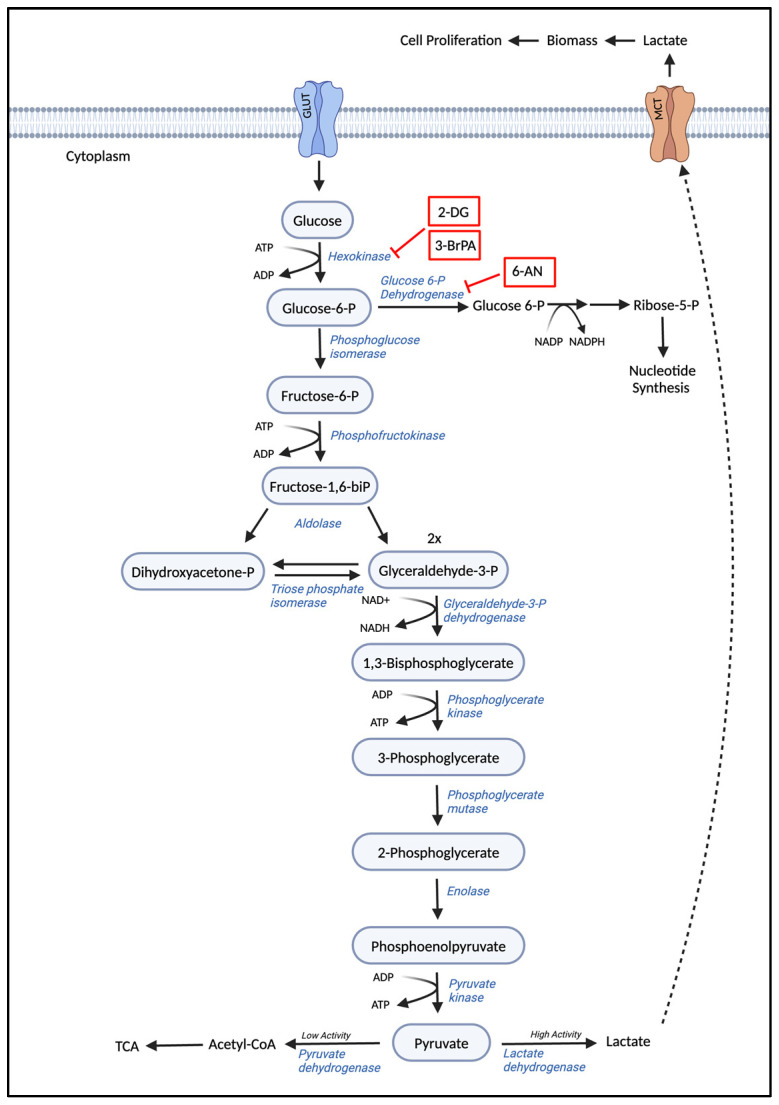
Glycolysis generates various metabolic intermediates that support cell growth and provide metabolites for secondary pathways. Glucose metabolism produces pyruvate, regardless of oxygen availability. In aerobic conditions, pyruvate enters the TCA cycle and participates in OXPHOS. In contrast, under low oxygen conditions, cells undergo anaerobic glycolysis, resulting in the production of lactate as a byproduct. AML cells heavily rely on glycolysis, making them particularly vulnerable to treatment with glycolytic inhibitors. The glycolytic inhibitors are indicated in the red boxes. Figure created with BioRender.com.

**Figure 3 cancers-17-01355-f003:**
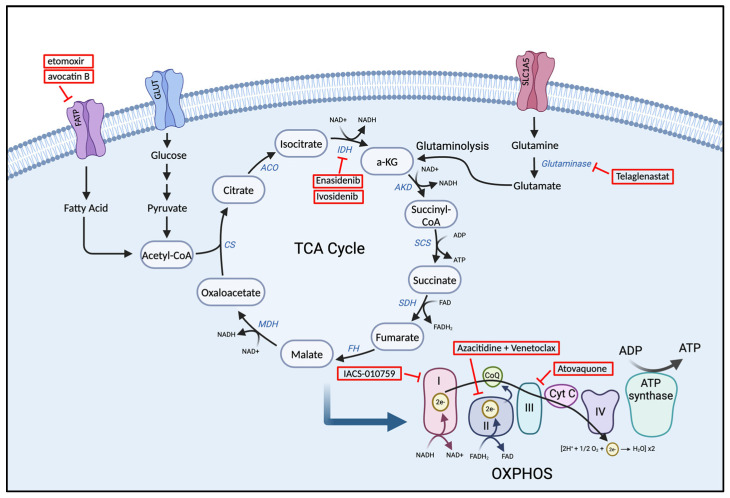
Energy metabolism in cells and metabolic inhibitors in use. During normal physiologic conditions, cells undergo glycolysis and produce pyruvate that is converted to acetyl-coA and enters the TCA cycle in the mitochondria. The TCA cycle produces substrates NADH AND FADH_2_ to the ETC. NADH supplies electrons to complex I while FADH_2_ to complex II. Transfer of electrons occurs from complex I through IV which allows the transfer of hydrogen protons into the inter mitochondrial membrane space. This process creates mitochondrial membrane potential allowing protons to enter mitochondria via complex V and generating ATP. Fatty acid β-oxidation also supplies acetyl-CoA to the TCA cycle. This figure illustrates the cellular energy metabolic pathway and the emerging treatments targeting metabolism in AML. (Shown inside the red boxes). Figure created with BioRender.com.

**Figure 4 cancers-17-01355-f004:**
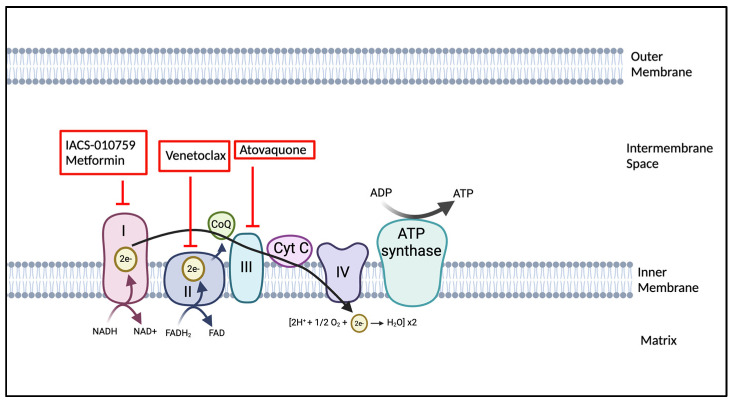
OXPHOS Inhibitors. Figure 4 illustrates the electron transport chain, which involves the transfer of electrons through a series of oxidoreduction reactions occurring in the inner mitochondrial membrane. This metabolic pathway, known as OXPHOS, generates a proton gradient as electrons flow through complex I (I), complex II (II), Coenzyme Q10 (CoQ), complex III (III), cytochrome c (Cyt C), and complex IV (IV), with O_2_ serving as the terminal electron acceptor. The electron transport chain is coupled with ATP synthase, which phosphorylates ADP to ATP by utilizing the proton gradient. Currently studied OXPHOS inhibitors targeting each complex are highlighted in red boxes. Figure created with BioRender.com.

**Table 1 cancers-17-01355-t001:** Emerging therapeutic compounds targeting metabolic pathways in AML.

Drug	Metabolic Target/Effect	Mechanism of Action	Stage of Development/NCT Number	Ref.
2-Deoxy-D-glucose (2-DG)	Glycolysis	Inhibition of hexokinase 2	Pre-clinical (in vitro and in vivo)	[144]
3-bromopyruvate (3BrPA)	Glycolysis	Inhibition of hexokinase 2	Pre-clinical (in vitro and in vivo)	[145]
IACS-010759	OXPHOS	Inhibition ETC1 activity	Phase I (NCT02882321)	[152]
Telaglenastat (CB-839)	Glutaminolysis	Inhibited GLS1 and glutathione	Phase I (NCT02071927)	[153,154]
BCT-100	Arginine metabolism	Arginine depletion	Phase I/II (NCT03455140)	[116]
ADI-PEG 20	Arginine metabolism	Arginine depletion	Phase II trial (NCT01910012)	[155]
Venetoclax	OXPHOS	BCL-2 inhibition	Phase I/II/III (NCT02993523, NCT04801797, NCT05177731, NCT05048615,NCT03586609)	[156,157]
LCL204	Lipid metabolism	Acid ceramide inhibition	Pre-clinical (in vitro and in vivo)	[139]
Tigecyclin	OXPHOS	Mitochondrial translation inhibition	Phase I (NCT01332786)	[158]
Ivosidenib (AG-120)	Genetic mutations	Inhibitor of mutant IDH1	Phase I/II/III (NCT02074839, NCT03173248)	[22,159,160]
Olutasidenib (FT-2102)	Genetic mutations	Inhibitor of mutant IDH1	Phase I/II (NCT02719574)	[161]
Enasidenib (AG-221)	Genetic mutations	Inhibitor of mutant IDH2	Phase I/II/III (NCT02577406)	[162]
Atovaquone	OXPHOS	Upregulating the ISR pathway and downregulating oxidative phosphorylation	Pre-clinical (in vitro and in vivo) Feasibility Trial (NCT03568994)	[163]
Avocatin B, Etomoxir, and ST-1326	Lipid metabolsim	Altering fatty acid oxidation	Pre-clinical (in vitro and in vivo)	[71,128,150]

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
