# Peer review of "Understanding and Targeting Metabolic Vulnerabilities in Acute Myeloid Leukemia: An Updated Comprehensive Review"

_cancers, 2025, doi:10.3390/cancers17081355_

Round 1
Reviewer 1 Report
Comments and Suggestions for Authors
The current manuscript provides well-structured and up-to-date overview of metabolic vulnerabilities in AML, including glycolysis, amino acid metabolism, OXPHOS, and lipid metabolism. The manuscript follows a logical progression, however I believe with following amendments the manuscript fits well in the scope of the journal publications.
Suggestions:
1- Certain concepts, such as the oxidative phosphorylation reliance and Warburg effect, and glutamine metabolism, are described by several times across different sections. The authors could consider combining the overlapping outlined sections to maintain conciseness.
2- While the manuscript discusses metabolic therapies, it does not sufficiently address the potential toxicities, off-target effects, and metabolic adaptations that could limit clinical success. Expanding this discussion would provide a more balanced perspective.
3- AML is highly heterogeneous at both the genetic and metabolic levels, yet the manuscript does not explore how metabolic dependencies might differ across AML subtypes (e.g., FLT3-mutated vs. NPM1-mutated AML). Including such distinctions would enhance the precision medicine approach advocated in the review.
4- Advances in metabolomics, single-cell sequencing, and CRISPR-based metabolic screens could further improve our understanding of AML metabolic vulnerabilities. Consider incorporating a section on how these technologies could refine therapeutic targeting.
Comments on the Quality of English Language
Further proofreading could improve readability, as there are some grammatical and typographical errors.
Author Response
"Please see the attachment."

Reviewer 2 Report
Comments and Suggestions for Authors
This is a well-organized and thorough review of metabolic changes which occur in AML
Comments:
- The introduction is long and quite general regarding leukemia and it could be shortened to focus on metabolic derangements in AML.
- In lines 256 and 257, it is rare for AML to evolve to CML with a BCR/ABL mutation. Perhaps this is an error and some other mutation should be inserted.
- While the figures are illustrative, it would be of help to the reader's understanding to create tables which list the changes in glycolysis, amino acid metabolism, and lipid metabolism.
- In lines 390 to 392, what is meant by insufficient levels of pyruvate "being converting" to acetyl-CoA in AML cells.
- Would define anaplersosis for the general reader.
- In the therapeutic strategies section, would separate pre=clinical from clinical studies; e.g., has 3-bromopyruvate been used in patients or just in cell lines described? This is better defined in Table 1.
- In that section, would use updated medication names consistently (e.g., AG-221 and AG-120. Would also mention oltuasudenib has US FDA approval in AML.
Minor: is pAML pediatric AML?
Author Response
"Please see the attachment."

Reviewer 3 Report
Comments and Suggestions for Authors
This is a very interesting area and needs to be in the forefront of researchers in the field. However, the subject matter of the review is similar to the review by Mishra et al Metabolism in acute myeloid leukemia: mechanistic insights and therapeutic targets published in Blood in 2023. It was difficult to properly distinguish the novelty, and progression in the field, of this manuscript following my reading of the Blood paper.
The Introduction was general and similar to many other reviews and teaching materials on AML introductions. It is dense to read before the paragraph outlining the subject and aims of the review. The language could be made more concise.
The sections were numbered incorrectly; “Metabolic Characteristics of Normal Hematopoietic Cells and Leukemic Cells” and “Altered Metabolic Pathways in AML” were both labelled section 2.
The style of writing is to provide a series of statements with a reference rather than a description or example of the idea being discussed. This is evident on page 4 in the discussion of Paget's "Seed and Soil" hypothesis. There are many examples of others writing about this hypothesis in the tumour field. This manuscript missed the opportunity to use the hypothesis more fully to illustrate concepts that were being discusses. One is not more the wiser for this sentence. Another example where more detailed thoughts about the subject would improve the manuscript is in the first paragraph in section 5. The three challenges are stated but there are no examples highlighting the challenges. For example are there trials that have failed, inhibitors that have showed promise but failed to progress to demonstrate why minimising off-targets is challenging. Can one have a “significant potential challenge”? (When is a challenge not significant?)
A second example is on the section discussing FIS1 (p6 last paragraph). There are statements describing FIS1 but the discussion ends with a statement saying FIS1 is an important LSC biomarker – how and what does it mark – poor prognosis? Given the reference for this was 2014 – why has FIS1 not made progress or has it?
The structure of the review would benefit from more sections or subheadings. The figures seemed poorly positioned and could be aligned with the text more clearly. This makes the figures a bit of a lost opportunity. For example, Figure 3 legend fails to explain that the red boxes in the figure show the AML related treatments.
The section in this manuscript comparing metabolic characteristics of normal and leukemic cells provided a structure that would be useful to follow for the remainder whereby the changes between normal HSC and leukemic cells are directly contrasted. However, section 3 changed styles and felt like a different paper.
A consistent use of figures and tables (perhaps one covering all parts of each section within each section) would help connect the paper together. Each figure has a lot of detail but the detail critical for enhancing the understanding of readers of this manuscript is missing (description of red boxes should be in each legend).
Overall, this is review needs major rewriting to provide the readers with perspective and a critique of the field - not a list of the literature in the area.
Author Response
"Please see the attachment."
